# RETHINKING STYLE AND CONTENT DISENTANGLEMENT IN VARIATIONAL AUTOENCODERS

**Rui Shu, Shengjia Zhao, & Mykel J. Kochenderfer**
Stanford University
{ruishu,sjzhao,mykel}@stanford.edu

## ABSTRACT

A common test for whether a generative model learns disentangled representations is its ability to learn style and content as independent factors of variation on digit datasets. To achieve such disentanglement with variational autoencoders, the label information is often provided in either a fully-supervised or semi-supervised fashion. We show, however, that the variational objective is insufficient in explaining the observed style and content disentanglement. Furthermore, we present an empirical framework to systematically evaluate the disentanglement behavior of our models. We show that the encoder and decoder independently favor disentangled representations and that this tendency depends on the implicit regularization by stochastic gradient descent.

## 1 STYLE VS CONTENT DISENTANGLEMENT

Let $\mathcal{D} = \{x^{(i)}, y^{(i)}\}$ be a labeled dataset where $x \in \mathcal{X}$ is the image and $y \in \mathcal{Y}$ is the label. A variational autoencoder (Kingma & Welling, 2013) introduces the latent variable $z \in \mathcal{Z}$ and can be trained on labeled data using the variational lower bound

$$\mathcal{L}(p, q) = \mathbb{E}_{x,y \sim \mathcal{D}} \mathbb{E}_{q(z|x,y)} \ln \frac{p(x, y, z)}{q(z \mid x, y)}, \tag{1}$$

where $p(x, y, z) = p(y)p(z)p(x \mid y, z)$ denotes the generative model with unit Gaussian $p(z)$ and uniform categorical $p(y)$, and $q(z \mid x, y)$ denotes the inference model. Similar to Kingma et al. (2014), we use conditional Gaussians for $p(x \mid y, z)$ and $q(z \mid x, y)$, and parameterize both conditional distributions with neural networks. We refer to the underlying neural networks as the encoder $\omega_q(x, y) = \{\mu_q(x, y), \Sigma_q(x, y)\}$ and the decoder $\omega_p(y, z) = \{\mu_p(y, z), \Sigma_p(y, z)\}$.

Note that $\mu_p : \mathcal{Y} \times \mathcal{Z} \to \mathcal{X}$ takes a vector of continuous $z \sim \mathcal{N}(\mathbf{0}, \mathbf{I})$ and a discrete $y \sim \text{Cat}(\frac{1}{10}, \ldots, \frac{1}{10})$ as input and outputs an image $x$. For style and content disentanglement on digit datasets (Mathieu et al., 2016; Siddharth et al., 2017), we say that the decoder disentangles style and content when the following two conditions hold:

1. The latent variable $y$ is label-preserving. For fixed $y \in \mathcal{Y}$, $\mu_p(y, z)$ outputs images of the same digit class for all $z \in \mathcal{Z}$.

2. The latent variable $z$ is style-preserving. For fixed $z \in \mathcal{Z}$, $\mu_p(y, z)$ outputs images of the same style for all $y \in \mathcal{Y}$.

### 1.1 LABEL PRESERVATION

To see that the variational autoencoder objective encourages label preservation, note that the objective can be rewritten as

$$\mathcal{L} = \mathbb{E}_{x \sim p_{\mathcal{D}}(x)} \left[ \ln p(x) - D_{\text{KL}}(p_{\mathcal{D}}(y \mid x) q(z \mid x, y) \| p(y \mid x) p(z \mid x, y)) \right], \tag{2}$$

where $p_{\mathcal{D}}$ denotes the labeled data distribution. By interpreting $p_{\mathcal{D}}$ as part of the variational posterior, we see that minimization of the Kullback-Leibler divergence term encourages the generator's true posterior $p(y \mid x)$ to be a good classifier for the labeled data. In other words, the variational objective uses *posterior regularization* to train the generator to properly encode the label information into the latent variable $y$.

## 1.2 STYLE PRESERVATION

Since $y$ strictly encodes the label information, the image style can only be encoded in $z$. However, it is not clear that the model will learn this encoding in a style-preserving fashion. In fact, style preservation is equivalent to solving a 10-way *unsupervised* domain alignment problem: each digit class subset is a separate domain and the goal is find an alignment over the ten domains such that the mapping from any domain to another occurs in a style-preserving fashion. We now show that the variational autoencoder objective does not enforce style preservation.

**Proposition 1** *For fixed $p(z)$ and $p(y)$, let $p^*(x \mid y, z)$ be a generator that is style-preserving with corresponding true posterior $q^*$. If $p$ and $q$ are infinite-capacity models, then there exists $p'$ and $q'$ such that $\mathcal{L}(p^*, q^*) = \mathcal{L}(p', q')$ but $p'$ is not style-preserving.*

*Proof.* Let $Z$ be the random variable for $p(z)$. Consider a set of transformations $\{T_i : \mathcal{Z} \to \mathcal{Z}\}_{i \in \mathcal{Y}}$ that are distribution-preserving ($\forall i \in \mathcal{Y}, T_i(Z) = Z$) and distinct ($\forall i \neq j, \exists z \in \mathcal{Z}$ such that $T_i(z) \neq T_j(z)$). Let $p'(x \mid y, z) = p^*(x \mid T_y(z), z)$ and let $q'$ be the corresponding true posterior. It follows that $\mathcal{L}(p^*, q^*) = \mathcal{L}(p', q')$. Furthermore, since $\{T_i\}$ are distinct transformations, $\exists i, j \in \mathcal{Y}$ and $z \in \mathcal{Z}$ such that $p'(x \mid i, z)$ and $p'(x \mid j, z)$ output images of different styles. $\qquad \square$

To account for a constrained distribution family, we can consider a variant of Proposition 1 that restricts $p(z)$, $p(x \mid y, z)$ and $q(z \mid x, y)$ to the isotropic Gaussian family and choose $\{T_i\}$ to be a set of rigid transformation matrices. Figure 1 shows how label-dependent rotations easily convert a disentangled generator $p^*$ into an entangled generator $p'$ (not style-preserving) despite the two generators inducing the exact same density over the image space $\mathcal{X}$. In general, any learning algorithm based on minimizing $d(p_{\mathcal{D}}(x, y), p(x, y))$ according to any divergence $d(\cdot, \cdot)$ cannot disambiguate $p^*$ from $p'$. While this observation was independently made in Szabó et al. (2017), we specifically study why learned models preserve style despite the under-constrained objective.

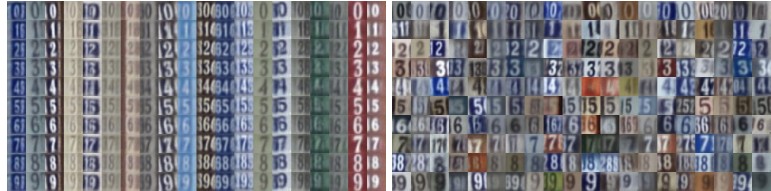

Figure 1: Left to right: Comparison of $\mu_p^*$ and $\mu_p'$ on SVHN. We sampled $z^{(1:20)}$. The image at row/column $(i, j)$ is the output of $\mu_p(i, z^{(j)})$. Note that at most one decoder can be style-preserving.

## 2 THE DISENTANGLEMENT BIAS IN SHARED ENCODERS AND DECODERS

Given the ease of constructing generators that are not style-preserving and the flexibility of neural networks, it is surprising that variational autoencoders consistently learn generators that are label *and* style-preserving. Furthermore, it is not obvious if this disentanglement bias arises from the shared encoder, decoder, or both.

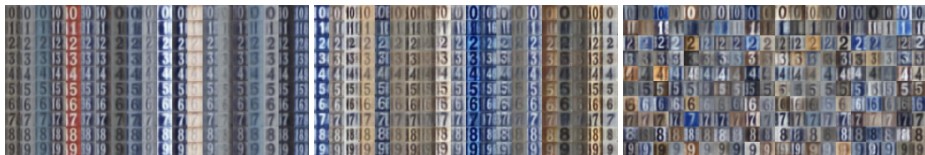

Figure 2: Left to right: output of $\mu_p$ when (1) $\omega_q$ is split, (2) $\omega_p$ is split, (3) $(\omega_p, \omega_q)$ is split.

To address this question, we exploit an alternative parameterization of the generative and inference models where the disentanglement bias is absent. Consider ten independent neural networks $\{\omega_p^{(i)}, \omega_q^{(i)}\}_{i \in \mathcal{Y}}$. Let $\omega_p(y, z) = \omega_p^{(y)}(y, z)$ and $\omega_q(x, y) = \omega_q^{(y)}(x, y)$. We shall refer to $\omega$ as *split*

since it is composed of ten separate neural networks—one for each digit class. Note that training on the split model $(\omega_p, \omega_q)$ is equivalent to training a separate variational autoencoder on each digit class subset. As shown in Figure 2, since the learned mapping between $z$ and $x$ is non-unique, the independent training of ten variational autoencoders without shared parameters is unlikely to achieve style alignment/preservation. By selectively splitting either the encoder or the decoder, we can thus evaluate the style preservation bias of its counterpart. Figure 2 shows that a shared encoder *or* shared decoder is sufficient for achieving disentanglement.

## 3   IMPLICIT REGULARIZATION BY STOCHASTIC GRADIENT DESCENT

Since the disentanglement bias cannot be attributed to the objective function for infinite-capacity models, we wish to determine whether it can be explained by the finite model capacity of our neural networks. Considering Figure 1, for example, the entangled decoder $\omega'_p$ might not be present in the hypothesis space of the shared decoder. Inspired by Zhang et al. (2016), we evaluate whether a shared encoder or decoder is capable of learning an entangled representation.

We first train a variational autoencoder to learn a disentangled inference model $q^*(z \mid x, y)$. We then sample $z \sim q^*(z \mid x, y)$ from the inference model to construct a dataset $\{x^{(i)}, y^{(i)}, z^{(i)}\}$. Using a set of label-dependent rotations $\{T_i\}$, we now train an encoder on the disentangled regression problem $(x, y) \mapsto z$ and the entangled regression problem $(x, y) \mapsto T_y(z)$. Analogously, we trained a decoder on $(y, z) \mapsto x$ and $(y, T_y(z)) \mapsto x$.

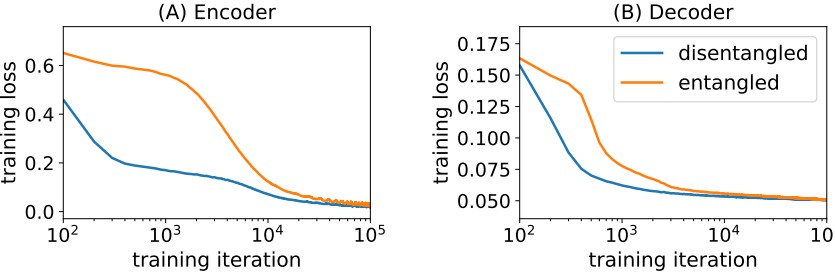

Figure 3: Comparison of the convergence rate and final loss of models trained on the disentangled versus entangled regression problem.

Figure 3 shows that the shared encoder and decoder converge to similar losses on both the entangled and disentangled regression problem, demonstrating that both models have sufficient capacity to learn an entangled representation. However, the convergence rate is significantly faster for the disentangled regression problem. This suggests that stochastic gradient descent preferentially converges to models with disentangled representations, although why convergence is faster for disentangled representations remains an open question.

## 4   DISCUSSION

We considered a definition of style and content disentanglement that decomposes disentanglement into style and label preservation. We then presented an experimental framework to evaluate the disentanglement behavior in variational autoencoders and showed both theoretically and empirically that the variational objective is insufficient in explaining the style-preserving behavior of the learned model. Our work raises the conceptually challenging question of why our models consistently achieve style and content disentanglement. It is tempting to consider the neural network's bias for style preservation in the representation space as the *deep representation prior* analog of the deep image prior (Ulyanov et al., 2017). Although we identify model parameterization and stochastic gradient descent as critical pieces of the puzzle, a complete explanation of disentanglement remains elusive and is of high theoretical and practical interest. As future work, our experimental framework can also be applied to analyze generative models that exhibit motion/pose and content disentanglement in videos.

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

## A WEAKENING THE DISENTANGLEMENT BIAS OF ENCODERS AND DECODERS

We present additional experimental results that complement the main findings of the paper. We first consider whether splitting a single layer of the shared encoder $\omega_p$ or shared decoder $\omega_p$ is sufficient to break its style preservation bias. We show in Figure 4 that when the split occurs close to the representation layer, style preservation is lost. Interestingly, in both the encoder and decoder, we observe that the further the split occurs from the representation layer, the more likely some degree of style preservation occurs.

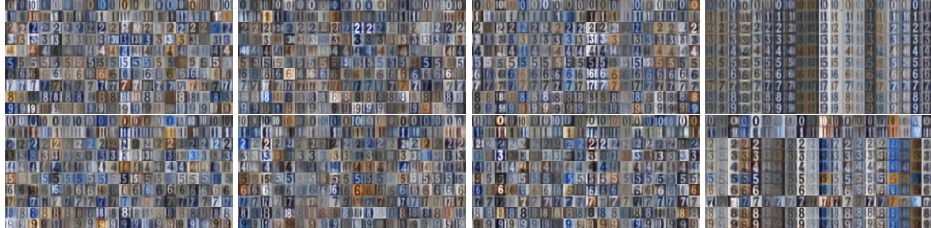

Figure 4: **Top row**. Encoder $\omega_q$ is always split. Denote $l$ as the decoder $\omega_p$'s first layer. Left to right: output when the layer $\{l, \dots, l+3\}$ is also split. **Bottom row**. Decoder $\omega_p$ is always split. Denote $l$ as the encoder $\omega_q$'s last layer . Left to right: output when the layer $\{l, \dots, l-3\}$ is also split.

# B  ENCODER VS DECODER DISENTANGLEMENT BIAS

Since the disentanglement of $\mu_p$ and $\mu'_p$ are mutually exclusive (Fig. 1), we can stress-test the style preservation bias of the encoder versus the decoder by using a training procedure that encodes the image with $\omega_q$, transforms the encoding with a label-dependent rotation, and then decodes with $\omega_p$. Thus, if the decoder $\omega_p$ learns a disentangled representation then the encoder $\omega_q$ and the transformed decoder $\omega'_p$ cannot, and vice versa. Figure 5 shows that the encoder has a stronger style preservation bias than the decoder when the representation space is subjected to label-dependent rotation. Interestingly, we also show that if even one of the encoder's layers is split, the decoder's bias dominates. The fact that the shared decoder can learn an entangled representation suggests that its bias for style preservation is not attributable to limited model capacity.

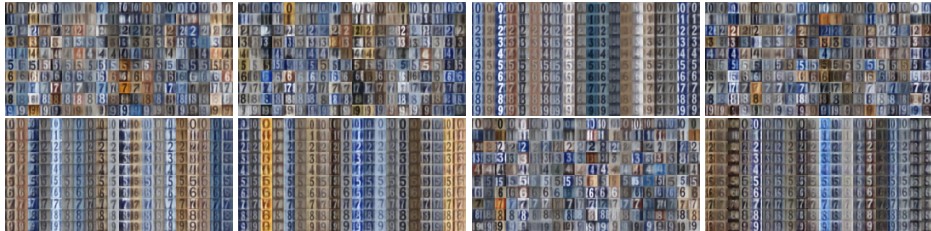

Figure 5: Label-dependent rotation. **Top row**. Left to right: output of $\mu_p$ (1) under standard competition conditions (encoder wins), (2) when $y$ is masked from the encoder $\omega_q$ (encoder wins), (3) when $\omega_q$'s layer $l-3$ is split (decoder wins), (4) when $\omega_p$'s layer $l+3$ is split (encoder wins). **Bottom row**. Corresponding output of $\mu'_p$. The encoder wins if the bottom row is disentangled, else the decoder wins.

We can conduct the same encoder versus decoder experiment, but with a reflection matrix instead of a rotation matrix. In particular, we chose the label-dependent reflection $T_i = (-1)^{\mathbb{1}\{i<5\}}$. Unlike the rotation experiment, the encoder's style preservation bias is weaker than the decoder's when the encoder observes $y$ as input. However, when $y$ is masked out, the encoder's style preservation bias is stronger.

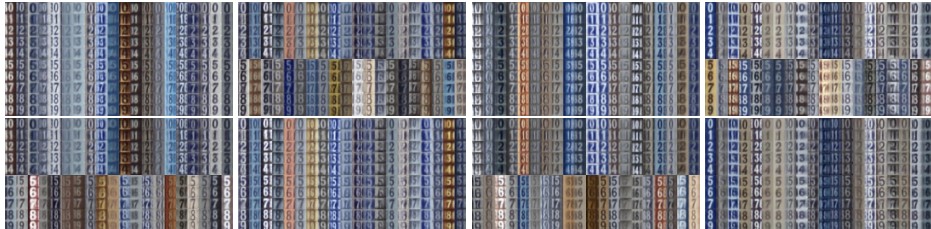

Figure 6: Label-dependent reflection. **Top row**. Left to right: output of $\mu_p$ (1) under standard competition conditions (decoder wins), (2) when $y$ is masked from the encoder $\omega_q$ (encoder wins), (3) when $\omega_q$'s layer $l-3$ is split (decoder wins), (4) when $\omega_p$'s layer $l+3$ is split (encoder wins). **Bottom row**. Corresponding output of $\mu'_p$. The encoder wins if the bottom row is disentangled, else the decoder wins.

In both the rotation and reflection competition experiments, we observe that the opposing model wins with non-zero probability under the standard competition setting. The exact win rate remains to be determined.

