# OpenReview forum: "Rethinking Style and Content Disentanglement in Variational Autoencoders"
_ICLR.cc/2018/Workshop — Accept_

### Official Review · AnonReviewer2 · 2018-03-10

**Rating:** 7
**Confidence:** 3

**Review:**

The paper addresses the problem of learning representations that disentangle style from label information. A simple but important observation is made: the loss function used for training VAEs does not promote disentanglement, despite the obtained results. The authors empirically show the importance of the inductive bias provided by the shared parameters in the decoder or decoder for obtaining these results.

The paper is well written and provides several convincing experiments on a simple dataset.

Please provide details of the architectures used. Are the encoders/decoders used convolutional or MLPs? How would that affect the results?

---

### Official Review · AnonReviewer3 · 2018-03-12
**this paper studies the relationship between disentangling, SGD, loss and bias in architectures**

**Rating:** 7
**Confidence:** 3

**Review:**

- This paper shows an interesting set of experiments investigating the causal relationship between disentangling and (SGD, loss). The claim is that regularization due to SGD is the main factor for disentanglement

- If i am not mistaken, similar types of conclusion were shown in the beta VAE work (Higgins et al), where they show that disentangling is mostly due to the optimization process. It's good to see more studies of it and the paper should better reflect such prior works.

- If its the optimization or inference is causing this, then training the neural net with other algorithms such as SPSA or other evolutionary methods is a good control test

---

### Official Review · AnonReviewer1 · 2018-03-16
**A submission that provides some insights, but lacks some (very recent) references**

**Rating:** 7
**Confidence:** 5

**Review:**

The authors explore the question why semi-supervised variational autoencoders that combine a discrete variable y with continuous variables z are able to disentangle style and content. They show that the variational objective admits solutions that have the same ELBO, but do not disentangle style and content. They look at architectures in contain separate encoders/decoders for each value y=k as a means of investigating which component of the model induces disentanglement. They additionally find some evidence to suggest that gradient descent converges more quickly towards an disentangled solution than an entangled one.

This is overall a very reasonable workshop contribution. My main comment would be that there are some (very recent) references missing from the discussion. These references point to the fact that we can decompose the ELBO into a number of terms, one of which takes the form of a total correlation (a higher-dimensional generalization of the mutual information) between latent variables under the marginal distribution

    q(y, z) = 1/N Σ_n q(y,z | x^n)

Based on this, a number of papers have emerged in which authors point out that we can successfully disentangle representations by emphasizing this total correlation term in the objective. This suggests, that, contrary to what the authors write in the abstract, the VAE objective *does* in fact contain a component that induces disentangled. More precisely put, in these models the prior  p(y, z) = p(y) p(z) factorizes. The VAE objective implicitly minimizes KL(q(y, z) || p(y) p(z)), which means that we should in principle learn a marginal distribution q(y, z) = q(y) q(z) that also factorizes, which implies the invariance conditions that the authors pose in section 1.

This aside, I see nothing wrong with this submission, and I am happy for it to appear, as long as the authors incorporate the references below, and perhaps include some discussion thereof.

References

1. Hoffman, M. D. & Johnson, M. J. Elbo surgery: yet another way to carve up the variational evidence lower bound. in Workshop in Advances in Approximate Bayesian Inference, NIPS (2016).

2. Kim, H. & Mnih, A. Disentangling by factorising. arXiv preprint arXiv:1802.05983 (2018).

3. Chen, T. Q., Li, X., Grosse, R. & Duvenaud, D. Isolating Sources of Disentanglement in Variational Autoencoders. arXiv:1802.04942 [cs, stat] (2018).

4. Gao, S., Brekelmans, R., Steeg, G. V. & Galstyan, A. Auto-Encoding Total Correlation Explanation. arXiv:1802.05822 [cs, stat] (2018).

---

> ### Public Comment · ~Rui_Shu1 · 2018-03-21
> **response**
>
> Thank you for the review. While unstated in this workshop paper, one of our goals is to critique the belief that a factorized prior implies disentanglement.
>
> I would therefore like to inquire about your statement that learning "a marginal distribution q(y, z) = q(y) q(z) that factorizes [...] implies the invariance conditions."
>
> Consider the proposal distribution q'(z, y | x) and entangled generator p'(x | y, z) described in Proposition 1. For simplicity, consider defining q'(x) as simply the density over x described by the joint model p'(x, y, z) = p'(x | z)p(z)p(y). Since q'(z, y | x) is constructed to be the true posterior for p'(x, y, z), and since we've defined q'(x) := p'(x), it is trivially true that
>
> q'(y, z) = q'(z)q'(y) = p(z)p(y),
>
> where q'(y, z) = int q'(x)q'(y, z | x) dx.
>
> However, by construction, the generative model p' entangles style and content. This provides the counter-example, showing that there exists some dataset (i.e. p_data(x) := q'(x)) for which factorization of q'(y, z) does not imply disentanglement (as defined in our paper).
>
> We appreciate any feedback on flaws in this line of reasoning.

---

> > ### Comment · AnonReviewer1 · 2018-03-22
> > **Thanks for the response**
> >
> > It is indeed true the factorization in itself does not guarantee disentanglement, depending on what definition of disentanglement one employs – and I should perhaps have expressed myself more carefully there. In practice however, there is strong evidence that factorization does help induce disentanglement, even if it is in itself not a sufficient condition.
> >
> > I also agree with the authors that the question of generating images with the "same" style z for different values of y is complicated. My understanding of your Proposition 1 is that you could have a VAE in which the marginals p(x) and q(x) are identical, as are the marginals p(y, z) and q(y, z), but does not associate the same style for a given value of z across different values of y. I think this observation is correct.
> >
> > The reason why I put the "same" in quotes here is that I am perhaps not entirely convinced that a notion of same-ness is always well-defined. Clearly, for a dataset like MNIST there are 10 digit classes and it is easy to detect violation of invariance conditions with respect to y. For the style variables z, there are features such as stroke thickness and slant, for which invariance would be comparatively obvious upon inspection. On the other hand, for a dimension in z that characterizes variation in  2's (say, more or less "curly"), it is not less clear what the corresponding "same" style should be for, say, a 7. As such, I'm somewhat inclined to accept any model in which y and z are independent as a valid disentangled representation (which differs from the definition that the authors employ here).
> >
> > Where I absolutely agree with the authors is that, in practice, the degree of parameter sharing in the encoder decoder framework plays a big role. In particular, I think that the authors make a very useful observation in noting that there is certainly no way to ensure that z values correspond to the "same" when we have encoders q(z | y=k, x, φ_k) and p(x | y=k, z,  θ_k) with distinct parameterizations θ_k and φ_k.

---

> > > ### Public Comment · ~Rui_Shu1 · 2018-03-22
> > > **Very much in agreement**
> > >
> > > Thank you, we really appreciated your feedback.
> > >
> > > Regarding what constitutes the "same" style and whether "style"+"content" preservation is admissible as a legitimate definition of disentanglement, we agree that such a notion of "sameness" of style is often times ill-defined. To give an even more extreme example: in CIFAR-10, it is very hard to intuit what it would mean for a plane and a cat to have the same style. The main reason we chose to experiment on the SVHN dataset is because of the numerous visual cues (color of background, foreground, thickness, rotation, serif/non-serif, etc) present that allow for easy human consensus on what constitutes a "sameness" of style.
> > >
> > > We also agree that the literature has observed factorization to help with inducing disentanglement. We believe, however, that more care needs to be taken in determining what we know/do not know. For example: although the ablation test of (VAE objective + deep net + SGD) versus (VAE objective + deep net + SGD + factorization regularization) may confirm that adding factorization regularization is, under certain circumstances, casually related to disentanglement, our workshop paper exposes that a proper theory of why factorization seems to improve disentanglement remains elusive.
> > >
> > > To end on a positive note, we believe this gap between our empirical observations and our theoretical understanding of what is and what causes disentanglement is exciting and worthy of careful consideration as a line of research.

---

### Decision · Program_Chairs · 2018-03-20
**ICLR 2018 Workshop Acceptance Decision**

**Decision:**

Accept

**Comment:**

Congratulations, your paper was accepted to the ICLR workshop.